# Increased Expression of Cell Surface SSEA-1 is Closely Associated with Naïve-Like Conversion from Human Deciduous Teeth Dental Pulp Cells-Derived iPS Cells

**DOI:** 10.3390/ijms20071651

**Published:** 2019-04-03

**Authors:** Emi Inada, Issei Saitoh, Naoko Kubota, Yoko Iwase, Tomoya Murakami, Tadashi Sawami, Youichi Yamasaki, Masahiro Sato

**Affiliations:** 1Department of Pediatric Dentistry, Graduate School of Medical and Dental Sciences, Kagoshima University, Kagoshima 890-8544, Japan; inada@dent.kagoshima-u.ac.jp (E.I.); kubonao@dent.kagoshima-u.ac.jp (N.K.); yamasaki@dent.kagoshima-u.ac.jp (Y.Y.); 2Division of Pediatric Dentistry, Graduate School of Medical and Dental Science, Niigata University, Niigata 951-8514, Japan; isaito@dent.niigata-u.ac.jp (I.S.); iwase@dent.niigata-u.ac.jp (Y.I.); murakami@dent.niigata-u.ac.jp (T.M.); sawami@dent.niigata-u.ac.jp (T.S.); 3Section of Gene Expression Regulation, Frontier Science Research Center, Kagoshima University, Kagoshima 890-8544, Japan

**Keywords:** naïve stem cells (NSCs), stage-specific embryonic antigen 1 (SSEA-1), human deciduous teeth dental pulp cells, induced pluripotent stem cells (iPSCs), cell-surface marker, epiblast stem cells (EpiSCs), grand-state cells, α1,3-fucosyltransferase IX gene (*FUT9*), reprograming-related drugs

## Abstract

Stage-specific embryonic antigen 1 (SSEA-1) is an antigenic epitope (also called CD15 antigen) defined as a Lewis X carbohydrate structure and known to be expressed in murine embryonal carcinoma cells, mouse embryonic stem cells (ESCs), and murine and human germ cells, but not human ESCs/induced pluripotent stem cells (iPSCs). It is produced by α1,3-fucosyltransferase IX gene (*FUT9*), and F9 ECCs having a disrupted *FUT9* locus by gene targeting are reported to exhibit loss of SSEA-1 expression on their cell surface. Mouse ESCs are pluripotent cells and therefore known as “naïve stem cells (NSCs).” In contrast, human ESCs/iPSCs are thought to be epiblast stem cells (EpiSCs) that are slightly more differentiated than NSCs. Recently, it has been demonstrated that treatment of EpiSCs with several reprograming-related drugs can convert EpiSCs to cells similar to NSCs, which led us to speculate that SSEA-1 may have been expressed in these NSC-like EpiSCs. Immunocytochemical staining of these cells with anti-SSEA-1 revealed increased expression of this epitope. RT-PCR analysis also confirmed increased expression of *FUT9* transcripts as well as other stemness-related transcripts such as *REX-1* (*ZFP42*). These results suggest that SSEA-1 can be an excellent marker for human NSCs.

## 1. Introduction

Due to the multipotential property of induced pluripotent stem cells (iPSCs), use of these iPSCs and their differentiated derivatives, with respect to regeneration of oral and maxillofacial tissues, is thought to be an important approach in the field of tooth regeneration [1]. For example, Xie et al. [2] tested the potential of human iPSCs generated from dental pulp stem cells to differentiate into functional odontoblasts through subcutaneous implantation of the dentin discs with poly-L-lactic acid scaffolds containing iPSCs into immunodeficient mice and found that the grafted iPSCs generated a pulp-like tissue with tubular dentin. Cai et al. [3] describe the generation of tooth-like structures from integration-free human urine-induced pluripotent stem cells (ifhU-iPSCs). They first differentiated ifhU-iPSCs to epithelial sheets, which were then recombined with the mouse embryonic dental mesenchyme before transplantation into the mouse subrenal capsule. They observed the presence of tooth-like structures in three weeks after grafting [3].

Notably, presently available human iPSCs are generally considered to be in a state called “epiblast stem cells (EpiSCs)” that are slightly more differentiated than pluripotential embryonic stem cells (ESCs) called “naïve stem cells (NSCs) or grand-state cells,” as exemplified by murine ESCs [4]. In this context, human iPSCs have the potential to differentiate into several types of specialized cells that are derived from the three germ layers, as mentioned above, but they are still not the perfect cell model for regeneration. If human iPSCs are successfully converted to NSCs, the resultant cells could be used for various studies in addition to regenerative medicine in the dental field.

Attempts to convert EpiSCs to NSC-like cells have been made by several laboratories. For example, Hana et al. [5] demonstrated that treatment of EpiSCs with reprogramming-related drugs such as 2i [PD0325901 (an inhibitor of the MEK/ERK pathway) + CHIR99021 (an activator of the Wnt/β-catenin signaling pathway by inhibiting the activity of glycogen synthase kinase 3)], kenpaullone (an inhibitor of glycogen synthase kinase 3), and forskolin (known to increase intracellular levels of cyclic AMP) resulted in successful conversion of EpiSCs into NSC-like cells. They demonstrated that the resulting NSC-like cells resembled the property of murine ESCs, as exemplified by the dome-like colony morphology, rapid proliferation, higher survival rate after trypsinization (which means lower dependency on the Rho-associated coiled-coil forming kinase (ROCK) inhibitor), and activation of both X-chromosomes in female NSCs [5]. Murayama et al. [6] demonstrated that EpiSCs could be converted into NSC-like cells when they are incubated in N2B27 medium containing small-molecule inhibitors IWP-2 (an inhibitor of Wnt production) or XAV939 (an inhibitor of the Wnt/β-catenin pathway) for 14 days. Unfortunately, little is known about the presence of useful cell-surface markers specifically recognizing NSCs. If such a molecule is identified, it should become possible to collect large amounts of NSCs by fluorescence-activated cell sorting (FACS), which will help accelerate basic research in the field of regenerative medicine employing iPSCs.

SSEA-1 (stage-specific embryonic antigen-1, CD15) is an antigenic epitope [Lewis X: Galβ1-4(Fucα1-3)GlcNAc] that is recognized by a monoclonal antibody (mAb) raised against fixed murine F9 embryonal carcinoma cells (ECCs) that are nullipotent [7]. This epitope is expressed in early mouse embryos, and highly undifferentiated cells such as NSC-type of murine ESCs/iPSCs, and germ cells such as primordial germ cells [8,9]. It is not expressed in human ESCs/iPSCs [10]. However, in adult humans it is shown to be expressed in neuronal stem cells [11], Paneth cells, intestinal stem cells [12], and stem cells from deciduous teeth membrane [13], suggesting that a gene coding for a protein involved in the synthesis of an antigenic epitope recognized by anti-SSEA-1 is not completely inactivated in humans. Notably, the antigenic epitope recognized by anti-SSEA-1 is known to be produced by the α1,3-fucosyltransferase IX gene (*FUT9*) [14]. Experiments using null mice, in which the *FUT9* locus had been completely disrupted by gene targeting, demonstrated that the knockout early embryos lacked expression of SSEA-1, but their offspring were normally born and viable with normal reproductive function, suggesting that SSEA-1 is not essential for embryonic development [14].

Recent advances in the reprogramming of somatic cells made it possible to create porcine iPSCs after the introduction of Yamanaka factors [15]. Almost all of these established porcine iPSCs lacked expression of SSEA-1, as in human ESCs/iPSCs [16]. However, Rodríguez et al. demonstrated that some iPS colonies exhibited SSEA-1 when immunocytochemical staining using anti-SSEA-1 was performed, although their staining was limited to some portion of a colony [17]. Unfortunately, they did not discuss the significance of the expression of SSEA-1 in the SSEA-1-positive porcine iPS colonies. Since SSEA-1 expression is linked to mouse ESCs/iPSCs that are known as NSCs, we speculated that these SSEA-1-positive porcine iPSCs are in the state of NSCs.

In this study, we examined whether human iPSCs, derived from human deciduous tooth dental pulp cells (HDDPCs) [18], begin to express SSEA-1 molecules when they are induced to convert to NSCs.

## 2. Results

### 2.1. Generation of HDDPC-Derived Naïve iPSCs

The addition of a cocktail (2i + kenpaullone + forskolin) to culture medium can support naïve characteristics of human iPSCs [5]. In order to convert EpiSC to NSC, EpiSCs (HDDPC-derived iPSCs) [18] were cultivated in NSC medium containing 2i (PD0325901 + CHIR99021) in a 60-mm dish containing mouse embryonic fibroblast (MEF)-derived feeder cells. As a control, EpiSCs were cultivated in a general medium called EpiSC medium. Medium change was performed every day by exchanging half of the medium with fresh medium. Cell passage was performed on the fifth day after cell seeding. No morphological alteration was noted when EpiSCs were cultured in NSC medium during the period after the first passage, but they exhibited NSC-like morphology, as exemplified by dome-like colonies (with an efficiency of ~10%; Figure 1A-a,b), within 4 days after the second passage and subsequent cultivation in NSC medium. On the other hand, EpiSCs cultivated in EpiSC medium remained as flat-shaped colonies (Figure 1A-c,d). These NSC-like colonies increased dramatically after the third passage. About 60% of the colonies (12/20 examined) showed dome-like morphology. Observation using confocal laser scanning microscopy also revealed that the height of each NSC-like colony was larger than that of EpiSC colonies (a vs. b in Figure 1B). Notably, the average diameter of each nucleus of the cells in the dome-like colonies, as evaluated by using Zeiss Cell Observer software, was significantly (*p* < 0.01) smaller than that of nuclei from the EpiSC colonies (Av. 11.4 vs. 13.2 μm; Figure 1C). We confirmed that there was no overlapping among 4′,6-diamidino-2-phenylindole (DAPI) -stained nuclei by measuring their diameter after preparation of digital images of individual nuclei, based on the 3D conversion software. The NSC-like colonies were also maintained stably after the fifth passage, but after the sixth passage, approximately 70% of the NSC-like colonies detached from the dish and formed an embryoid body-like structure with a cavity in their central portion. The remaining 30% stayed attached to the dish with a dome-like morphology.

### 2.2. Characterization of NSC-Like Colonies

To examine whether the resultant NSC-like colonies express pluripotency-related stemness factors at a molecular level, RT-PCR analysis was performed. Colonies (~100) corresponding to NSC and EpiSC (each harvested after the 3rd passage) were subjected to RNA isolation. RNAs derived from human cervical carcinoma cell line HeLa, human ovarian carcinoma cell line PA-1 and human skin fibroblasts HDFa were used as controls. Expression of reduced expression protein 1 (REX-1)/zinc finger protein 42 homolog (ZFP42), a zinc finger family transcription factor that is a well-known marker of NSC [19], was found to be predominant in NSC-like colonies (Figure 1D). mRNA encoding alkaline phosphatase (ALP) (corresponding to L/B/K ALP), a protein known to be expressed in undifferentiated cells such as iPS/ES cells [20], was also abundantly detectable in NSC-like colonies (Figure 1D). On the other hand, the expression of fibroblast growth factor-5 (FGF-5), a protein known to be predominantly expressed in EpiSCs [21,22], was only limited to EpiSCs (Figure 1D).

NSC has a greater potential to differentiate into various types of cells, including germ cells, than EpiSC [23]. To test this, we transplanted colonies (~300) corresponding to NSC and EpiSC (each harvested after the fifth passage) into pancreatic parenchyma of immunocompromised mice (BALB/c-*nu/nu*), as depicted in Figure 2A. This allows a small number of iPS colonies to grow in vivo, leading to the formation of a solid tumor, called teratoma [24]. Approximately, one month after cell inoculation, the mouse grafted with NSC-like colonies had a large solid tumor about 20 mm in diameter (enclosed by dotted lines in Figure 2B-a). In contrast, the mouse grafted with EpiSC colonies had tumors (enclosed by dotted lines in Figure 2B-b) after ~1.5 months after inoculation, but the tumor size appeared to be smaller than that derived from NSC-like colonies (a vs. b in Figure 2B). Histological examination of the resulting teratomas revealed that the teratoma derived from grafting of NSC-like colonies contained several types of differentiated cells including foci comprising of unknown cells (Figure 2C-a,b), ductal structure (Figure 2C-c,d), vessels with microvilli facing internal lumen (Figure 2C-e), eosinophilic structure (Figure 2C-f), and amorphous structure enriched with mesenchymal cells (Figure 2C-g,h). The EpiSC-derived teratoma also had differentiated cells such as ductal structure (Figure 2C-i,l), keratinocytes (Figure 2C-j) and adipocytes (Figure 2C-k). More importantly, the NSC-derived teratoma had some cell aggregates comprised of vasa homolog (MHV)-positive cells (arrows in Figure 2D-a,b), suggesting differentiation of germ cells in this tumor. In the EpiSC-derived teratoma, there was a portion that was positive for anti-vasa homolog (arrowhead in Figure 2D-c,d), but its structure appeared to be not well-organized. Some tubular structures resembling embryonic kidney cells were also positive for this antibody (boxes shown in the right bottom in Figure 2D-c,d).

The formation of embryoid bodies (EBs) can be a good indicator of the ability of ESCs/iPSCs to differentiate in vitro. Colonies corresponding to NSC and EpiSC (each harvested after the third passage) were seeded onto non-adherent dishes containing Dulbecco’s modified Eagle’s medium (DMEM)/10% fetal bovine serum (FBS) and cultured for 10 days. Both NSC-like colonies and EpiSC colonies formed EBs in vitro (Appendix A). The resulting aggregates were next transferred to an adherent tissue culture dish to see possible outgrowth of differentiated cells from the EBs. On ~10 days after seeding, various types of cells including the origin of three germ layer (endoderm, mesoderm, and ectoderm) were discernible for both NSC-like colonies and EpiSC colonies (Appendix A). Notably, it seemed that the NSC-derived outgrowth had more abundant types of cells than the EpiSC-derived one (b vs. g in Appendix A). These findings demonstrated that it is possible to obtain NSC-like cells with high differentiation ability by cultivation in NSC medium.

### 2.3. Immunocytochemical Staining of NSC-Like Colonies

Next, we performed immunocytochemical staining to determine whether the resultant NSC-like colonies could express pluripotent stem cell markers such as OCT3/4, NANOG, Tra-1-60, and SSEA-4. As expected, the NSC-like colonies (on 4 days after the third passage) were reactive to the antibodies produced against these molecules (Appendix A).

In mice, SSEA-1 is thought to be highly expressed in murine ESCs and nulli-potential ECCs F9 [25,26]. However, in humans, EpiSC is thought to be unreactive to anti-SSEA-1 [23]. We thus examined whether the NSC-like colonies, obtained by cultivation of EpiSCs in NSC medium, would express SSEA-1. NSC-like colonies generated 3 to 4 days after the fourth passage were stained with anti-SSEA-1. EpiSC colonies maintained in EpiSC medium were also subjected to staining with anti-SSEA-1. In this case, less than 20% of an entire surface of a colony stained by anti-SSEA-1 was judged as low (L) level expression of SSEA-1 (as exemplified by “L” in Figure 3A-a,b); 20 to 80% of an entire surface of a colony stained by anti-SSEA-1 was judged as medium (M) level expression for SSEA-1 (as exemplified by “M” in Figure 3A-c,d); and > 80% of an entire surface of a colony stained by anti-SSEA-1 is judged as high (H) level expression for SSEA-1 (as exemplified by “H” in Figure 3A-e,f). About 80% of EpiSCs were less reactive to anti-SSEA-1 (group “L” of EpiSC in Figure 3B). In contrast, low-level expression of SSEA-1 in NSC-like colonies was about 20% (group “M” of NSC in Figure 3B). Furthermore, about 50% of NSC-like colonies exhibited higher level expression of SSEA-1 (group “H” of NSC in Figure 3B), which is significantly higher than that (~5%) of EpiSCs (group “H” of NSC in Figure 3B). Notably, the NSC-like colonies generated after the fourth passage were also positively stained by another lot of anti-SSEA-1 (Appendix A), suggesting no significant difference between the two lots of anti-SSEA-1. As shown in Figure 3A-e,f, an entire surface of NSC-like colonies were positively stained by anti-SSEA-1. We next tested whether this positive staining pattern is also seen at a single cell level. When NSC-like colonies were dissociated into single cells or small cell aggregates by trypsinization and subsequent pipetting, these dissociated cells were found to be strongly stained by anti-SSEA-1 (arrows in Appendix A). In contrast, single cells or small aggregates derived from EpiSC colonies were not stained or less stained by anti-SSEA-1 (arrowheads in Appendix A). Next, to abolish the activity of anti-SSEA-1, NSC-like colonies were reacted with anti-SSEA-1 that had been preincubated with 1 mM fucose. As a result, no obvious staining was noted (a–c vs. d–f in Appendix A). rBC2LCN lectin exhibits a high affinity for H-type 3 (Fucα1-2Galβ1-3GalNAc), a mucin-like O-type sugar chain on podocalyxin present on the surface of human ESCs/iPSCs [27,28], and is used as an undifferentiated marker, like SSEA-1 [29]. Staining of the NSC-like colonies with fluorescence-labeled rBC2LCN resulted in positive staining with this lectin (Appendix A). This positive staining was also abolished by preincubation of fluorescence-labeled rBC2LCN in the presence of 1 mM fucose (Appendix A). These results demonstrate that NSC-like colonies indeed express specific carbohydrate epitope recognized by SSEA-1 and rBC2LCN on their surface.

To confirm whether expression of *FUT9*, one of several alpha-3-fucosyltransferases, that can catalyze the last step in the biosynthesis of Lewis antigen which is the addition of a fucose to precursor polysaccharides [30], is elevated in the NSC-like colonies, RT-PCR was performed. As shown in Figure 1D, *FUT9* mRNA was predominantly expressed in NSC-like colonies.

## 3. Discussion

There are several ways to obtain naïve iPSCs. One is reported as transfection of somatic cells with Yamanaka’s factors and subsequent cultivation under specific culture conditions [31] and the second method is to treat EpiSCs with chemicals such as a cocktail containing 2i + kenpaullone + forskolin [5], small molecule inhibitors (e.g., IWP-2) [6], a cocktail called 5i/L/AF [32] which contains 5i (PD0325901, CHIR99021, and SB590885, WH-4-023, and Y-27632) supplemented with leukemia inhibitory factor (LIF), basic fibroblast growth factor (bFGF) (also known as FGF2), and activin A, or cocktail called t2iLGö [33], which contains CHIR99021, PDO325901, bFGF, and Gö6983. More recently, it was demonstrated that naïve human ESCs or iPSCs could be established without transgene expression in the presence of LIF, bFGF, and activin A and inhibitors against four signaling pathways (ERK1/2, GSK3β, JNK, and p38) for stable cell propagation [34]. These methods appear to be useful for generating NSCs, but chemical induction is more convenient because of its simplicity.

In this study, we employed the method of Hana et al. [5] who used the cocktail containing 2i + kenpaullone + forskolin. When EpiSCs were maintained in the NSC medium, NSC-like colonies showing dome-like morphology appeared after the second passage. These NSC-like colonies maintained a stable shape up to the fifth passage. In the in vivo and in vitro experiments of this study, we found the following: (1) the NSC-like colonies expressed the OCT3/4, NANOG, Tra-1-60, and SSEA-4 (see Appendix A), all of which were also observed in the EpiSCs [18,35]; (2) both EpiSC and NSC-like colonies had the ability to differentiate into all three embryonic germ layers when they were induced to differentiate in vitro (see Appendix A); (3) the NSC-like colonies proliferated more rapidly than the EpiSC colonies when transplanted into the pancreas of nude mice (see Figure 2B), and the former had a wider variety of differentiated cells than the latter (see Figure 2C); (4) the teratoma derived from NSC-like (but not EpiSC) colonies had structures which were organized to some extent and comprised of vasa homolog-positive cells (see Figure 2D); and (5) the NSC-like colonies had preferential expression of *REX-1*, known as a marker for NSC (see Figure 1D). Furthermore, expression of *FGF-5*, a marker known to be preferentially expressed on EpiSCs [21,22], was remarkable in EpiSCs, but not in the NSC-like colonies (see Figure 1D). Based on these results, it can be concluded that EpiSCs obtained after culture in NSC medium are considered successfully converted to NSCs.

Notably, about 70% of the NSC-like colonies tended to detach from the dish when they were cultivated in NSC medium after the sixth passage. It remains unknown why such an event frequently occurs, given that we always seeded iPSCs on the feeder cells in a gelatin-coated dish to maintain them in vitro, which is a technique generally employed as part of the standard cultivation of iPSCs. Likely, the property that allows NSC-like colonies to attach to the feeder cells or substratum may have been modified during culture. Notably, in case of chemically induced bovine NSC-like cells, they could be successfully maintained, without visible alteration in colony morphology, over 10 generations in vitro [36]. Furthermore, the teratomas generated after grafting of NSC-like colonies (picked up after the fifth passage) into pancreatic parenchyma of nude mice contained a variety of differentiated cells including vasa homolog-positive cells (see Figure 2D), suggesting that the NSC-like colonies still maintained their pluripotency. Notably, we observed vasa homolog-positive tubular structure (resembling embryonic renal structure) in the specimens of EpiSC-derived teratoma as well (see the column on the right bottom of Figure 2D-c,d). It remains unknown whether anti-vasa homolog can react with embryonic renal cells.

Especially, the size of a nucleus of NSC-like cell converted from EpiSC was smaller than that of EpiSC, when images of DAPI-stained cells (taken by a confocal microscope) were analyzed using Zeiss Cell Observer software (see Figure 1B,C). This appeared to reflect the property of actively proliferating cells and is probably one of the properties distinguishing EpiSC from NSC. 

Currently, there are several known markers defining NSC, as exemplified by REX-1 [19,37], transcription factor CP2 like 1 (TECP2L1; a CP2 family transcription factor that plays an important role for maintaining the naïve state) [38], STELLA [a nuclear transcription factor involved in the development of primordial germ cells and also known as developmental pluripotency associated 3 (DPPA3)] [39], estrogen-related receptor beta (ESRRB; a nuclear transcription factor capable of inducing the expression of genes associated with pluripotency during reprogramming) [39], and T-Box 3 (TBX3; a nuclear transcription factor that controls the STELLA) [39]. All these markers exist inside the cells. To our knowledge, there is no cell-surface marker specific to NSC. As mentioned previously, SSEA-1 expression is known to occur in murine iPSCs/ESCs, but not in human iPSCs/ESCs. Here we found that SSEA-1 is expressed when human iPSCs are converted to naïve state, which was verified by immunostaining with anti-SSEA-1 (see Figure 3A) and by cytochemical staining using lectin rBC2LCN that is specific to the Fucα1-2Galβ1-3GalNAc epitope (see Appendix A). Increased expression of SSEA-1 is also found to be associated with increased mRNA expression for FUT9 (see Figure 1D), a key enzyme synthesizing SSEA-1 antigen epitope [14]. As mentioned in the earlier, human iPSCs and ESCs are generally believed to be negative for the expression of SSEA-1 [10,23]. However, the inability to express SSEA-1 is not ascribed to the dysfunction of FUT9 enzyme, since SSEA-1 is indeed expressed in some tissues/organs of adult humans [13]. Thus, it is reasonable to consider that FUT9 expression is suppressed in human iPSCs and ESCs. We found that NSC-like colonies between third to fourth passage after cultivation of EpiSC in NSC medium began to express SSEA-1 on their surface (see Figure 3A). Notably, among these cell populations, about 20% of colonies were found to be classified as L group as they exhibited SSEA-1 on less than 20% of the entire cell surface of the colony (see Figure 3B). This suggests that SSEA-1 expression occurs in a stepwise fashion. According to Rodríguez et al. [17] who created porcine iPSCs by transfecting plasmid constructs carrying Yamanaka’s factor genes into embryonic fibroblasts, some of the established iPS colonies expressed SSEA-1. Although the authors did not specify that the SSEA-1-positive cells are NSCs, it is possible that they are iPSCs in the naïve state. Since SSEA-1 is a cell-surface antigen, FACS-based cell sorting of SSEA-1-positive cells appear to be possible, which will help to concentrate NSCs from the conventional iPSCs that are thought to be comprised of various types of cells such as the primed and naïve type of cells.

In summary, we demonstrate that the chemical conversion of EpiSCs to NSC-like cells is possible when human HDDPC-derived EpiSCs are used as an initial source and found that SSEA-1 expression is evident on these NSC-like cells. SSEA-1-based screening should facilitate the enrichment of NSC-like cells by FACS, which will be useful for various fields of studies employing iPSC-based regenerative medicine.

## 4. Materials and Methods

### 4.1. Ethical Approval

Generation and cultivation of HDDPC-derived iPSCs were performed according to the guideline and the protocol approved by the Ethical Committee for the Use and Experimentation of the Kagoshima University Graduate School of Medical and Dental Science (No. 27-11; dated on 29 May, 2015). Furthermore, the experiments described here were performed in agreement with the guidelines of Kagoshima University Committee on Recombinant DNA Security (No. 25076; dated on 27 March, 2014) and with approval by the Animal Care and Experimentation Committee of Kagoshima University (No. MD14003; dated on 17 April, 2014).

### 4.2. Generation and Maintenance of HDDPC-Derived iPSCs

Before the generation of HDDPC-derived iPSCs, we examined the stemness marker expression of HDDPCs. We found that they expressed OCT4 and NANOG as well as ALP [40]. Thus, we considered that HDDPCs contain immature cells (probably stem cells), together with more differentiated cells such as fibroblasts [35].

The HDDPC-derived iPSCs were generated using our protocol with slight modifications. HDDPCs were reprogrammed into iPSCs via electroporation of three kinds of plasmids carrying reprogramming factors OCT4, SOX2, KLF4, and L-MYC. The electroporated cells were then seeded onto three wells of a gelatin-coated 24-well plate containing Dulbecco’s modified eagle medium (DMEM) supplemented with 20% FBS. Seven days after transfection, the cells were trypsinized and subsequently re-seeded onto mitomycin C (MMC) (#M4287; Sigma-Aldrich, St. Louis, MO, USA) -treated MEFs in a 60 mm gelatin-coated dish (#4010-020; Iwaki Glass Co. Ltd., Tokyo, Japan) in EpiSC medium, which is comprised of human ES cell culture medium iPSellon (#007001; Cardio, Kobe, Japan), 5 ng/mL recombinant human bFGF (#064-04541; Wako Pure Chemical Industries, Ltd., Osaka, Japan), and 0.01 μg/mL recombinant human LIF (#129-05601; Wako Pure Chemical Industries, Ltd.). Approximately two weeks later, small iPSC-like colonies emerged. Medium change was performed every day by replacing a half of the medium with fresh medium. Cell passage was performed by trypsinization on the fifth day after cell seeding.

### 4.3. Induction of NSC-Like iPSCs

The HDDPC-derived iPSCs were cultured in NSC medium, based on N2B27 medium containing DMEM/F12 (#12634-010; Thermo Fisher Scientific, Inc., Waltham, MA, USA), Neurobasal medium (#21103-049; Thermo Fisher Scientific, Inc.), N2 supplement (#17502-048; Invitrogen, Carlsbad, CA, USA), and B27 supplement (#17504044; Invitrogen) in a ratio of 48:48:1:2, respectively. In the NSC medium, 5 ng/mL recombinant human bFGF, 0.02 μg/mL recombinant human LIF, 1 μM PD0325901 (#162-25291; Wako Pure Chemical Industries, Ltd.), 10 μM PD98059 (#169-19211; Wako Pure Chemical Industries, Ltd.), 3 μM CHIR99021 (#038-23101; Wako Pure Chemical Industries, Ltd.), 10 μM forskolin (#067-02191; Wako Pure Chemical Industries, Ltd.), and 1 μM kenpaullon (#110-00831; Wako Pure Chemical Industries, Ltd.) were also included. Medium change and cell passage were done as for cultivation of EpiSCs, as mentioned above. 

### 4.4. In Vitro Differentiation

To induce EB formation, NSC-like and EpiSCs colonies were dissected mechanically into pieces using a pipette tip under a stereomicroscope and then seeded onto an ultralow attachment 30 mm dish (#MS-9035X; Sumitomo Bakelite Co., Ltd., Tokyo, Japan) with DMEM/10% FBS. Ten days after cultivation, emerging EBs were transferred onto a gelatin-coated 24-well plate (#3820-024; Iwaki Glass Co. Ltd.) and cultured for another 10 days in DMEM/10% FBS to allow enhanced differentiation into various cell types. These cells were then subjected to immunostaining with antibodies, as described below.

### 4.5. In Vivo Differentiation

For inducing solid tumor formation in vivo, NSC-like colonies (~300) or EpiSCs colonies (~300) were harvested by simple pipetting after the fifth passage and then grafted beneath the pancreatic parenchyma of nude mice (BALB/cAJcl-*nu/nu*; CLEA Japan Ltd., Tokyo, Japan), according to the method of Sato et al. [24]. One to 1.5 months post-transplantation, the emerging teratomas were harvested and fixed with 4% paraformaldehyde (PFA) at 4 °C for 1 week. The fixed tissues were then dehydrated by immersion in 0.25% sucrose in Dulbecco’s modified phosphate buffered saline without Ca^2+^ and Mg^2+^ [D-PBS(-)] at 4 °C for two days, and then dehydrated in 0.4% sucrose in PBS at 4 °C for four days. These samples were then embedded in optimum cutting temperature (O.C.T.) compound (Tissue-Tek^®^ [no. 4583]; Miles Scientific, Naperville, IL, USA) for cryostat sectioning. Some cryostat sections were stained with hematoxylin and eosin (H-E), and others were subjected to immunostaining with antibodies, as described below.

### 4.6. Observation of Colony Morphology

To compare the morphology of NSC-like and EpiSC colonies, nuclear staining was performed using 1.5 µg/mL 6-diamidino-2-phenylinodole (DAPI) (#H-1200; Vector Laboratories Inc., Burlingame, CA, USA) and observed with a confocal laser scanning microscope (ZEISS LSM 880 with Airscan; Carl Zeiss AG, Jena, Deutschland). In addition, 20 nuclei were randomly selected for the 3D conversion software-based analysis from each colony and the diameter of the cells was measured to compare the sizes of both cells using Zeiss Cell Observer software (ZEN; Carl Zeiss AG).

### 4.7. RT-PCR Analysis

Total RNA from ~ 100 iPSC colonies was isolated using an RNA mini kit (#50204; QIAGEN N.V., Venlo, Limburg, Nederland). To identify the expression of target mRNA by RT-PCR, RT was first performed using a first-strand cDNA synthesis kit (#18080-051; Invitrogen). The resulting cDNAs were then PCR-amplified from undiluted cDNA samples (1 µL) in a total volume of 20 μL using AmpliTaq Gold^®^ 360 Master Mix (#4398881; Applied Biosystems, Foster City, CA, USA). PCR was performed for 38 cycles of 30 s denaturation at 95 °C, 30 s annealing at 58 °C, and 60 s extension at 72 °C in a PC708 thermal cycler (Astec, Fukuoka, Japan). cDNAs isolated from HeLa, HDFa, and PA-1 were used as controls. The sense and antisense primers used are listed in Table 1. The PCR products (2 µL) were analyzed by 2% agarose gel electrophoresis and subsequent staining with ethidium bromide.

### 4.8. Immunocytochemical Staining Using Antibodies

Cultured cells and colonies were fixed with 4% PFA for over 1 h at 4 °C and washed three times with D-PBS(-). They were then subjected to permeabilization with 0.05% Triton X-100 (#T8787; Sigma-Aldrich) for 3 min at room temperature (about 25 °C). After washing with D-PBS(-) containing 1% normal goat serum (NGS) (Invitrogen) (hereafter referred to as PBS/NGS), cells were blocked with 20% AquaBlock tm/EIA/WB (#PP82; East Coast Biologics, Inc., North Berwick, ME, USA) for 30 min at 4 °C, prior to incubation with the primary antibodies. In case of immunostaining using cryostat sections, pre-treatment of the section was done in the same way as for staining of cells and colonies, except for the omission of the permeabilization procedure.

After washing with PBS/NGS, the specimens were stained at 4 °C overnight with the following primary antibodies: OCT3/4 (1:200) (#sc-9081; Santa Cruz Biotechnology, Dallas, TX, USA), NANOG (1:200) (#RCAB0004P; Repro Cell, Kanagawa, Japan), Tra-1-60 (1:500) (#MAB4360-20; Millipore-Chemicon, Darmstadt, Germany), SSEA-4 (1:500) (#MAB4304-20; Millipore-Chemicon), α-fetoprotein (AFP) (1:200) (#HPA010607; Atlas Antibodies, Stockholm, Sweden), βIII-tubulin (1:200) (#NB110-57611; Novus Biological, AL, USA), α-smooth muscle actin (α-SMA) (1:200) (#NB600-531; Novus Biological), SSEA-1 (1:500) (Lot no. GR38347-10 and GR287616-11; #ab16285; Abcam, Cambridge, UK), and anti-DDX4/MVH (1:200) (#ab13840; Abcam). After staining and washing, the specimens were next stained at 4 °C overnight with the following secondary antibodies: Alexa Fluor 594-conjugated goat anti-mouse IgM (1:200) (#A21044; Invitrogen), Alexa Fluor 488-conjugated goat anti-rabbit IgG H&L (1:200) (#ab150077; Abcam), Alexa Fluor 647-conjugated goat anti-rabbit IgG Fab2 (1:200) (#4414s; Cell Signaling Technology, Tokyo, Japan), and fluorescein isothiocyanate (FITC)-conjugated goat anti-mouse IgG (1:200) (#AP503F; Millipore-Chemicon). After washing with PBS/NGS, nuclear staining was performed using DAPI.

For negative staining in case of use of anti-SSEA-1, anti-SSEA-1 (1:1000) was preincubated with 1 mM fucose (#ABS-00006290; FUJIFILM, Tokyo, Japan) at a ratio of 1:1 (*v*/*v*) at 4 °C overnight before reaction with the specimens.

### 4.9. Lectin Cytochemistry

rBC2LCN specifically recognize a sugar chain which exists on the surface of human pluripotent stem cells. We used a FITC-labeled rBC2LCN (1:200) (#180-02991; Wako Pure Chemical Industries, Ltd.). The fixed specimens were incubated with PBS/NGS containing FITC-labeled rBC2LCN at 4 °C for >30 min. After washing with PBS/NGS, nuclear staining was performed using DAPI.

For negative staining using FITC-labeled rBC2LCN, FITC-labeled rBC2LCN (1:200) was preincubated with 1 mM fucose at a ratio of 1:1 (*v*/*v*) at 4 °C overnight before reaction with the specimens.

### 4.10. Fluorescence Observation

Fluorescence was examined using an Olympus BX60 fluorescence microscope. Microphotographs were obtained using a digital camera (FUJIX HC-300/OL; Fuji Film, Tokyo, Japan) attached to the fluorescence microscope and printed using a Mitsubishi digital color printer (CP700DSA; Mitsubishi, Tokyo, Japan).

### 4.11. Statistical Analysis

Student’s *t*-test was used to test for differences in the nuclear size between NSC-like and EpiSC colonies. Significance was set at *p* < 0.05. The expression levels of SSEA-1 in NSC-like and EpiSC colonies were evaluated using the Kruskal-Wallis test and Steel-Dwass as a post-hoc test with a significance level of 0.05.

## Figures and Tables

**Figure 1 ijms-20-01651-f001:**
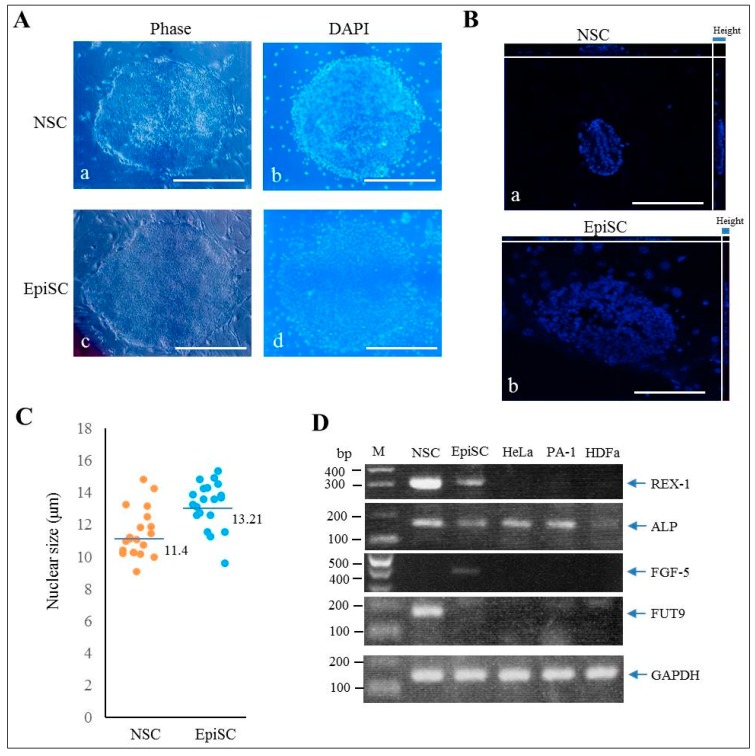
Characterization of HDDPC-derived naïve iPSCs. (**A**) Morphology of NSC-like colony (**a,b**) cultivated for 4 days in NSC medium after the fourth passage and EpiSC colony (**c**,**d**) continuously cultivated in EpiSC medium. Colonies were stained with DAPI after fixation. Phase, photos were taken under light; DAPI, photos were taken under UV illumination + light. Bar = 200 μm. (**B**) DAPI-derived fluorescence observation using a confocal laser scanning microscope. The image was analyzed using Zeiss Cell Observer software. The height of each colony is shown on the left side. Bar = 200 μm. (**C**) The nuclear size of each cell in an NSC-like or EpiSC colony determined using Zeiss Cell Observer software and plotted. Average of nuclear size is shown by bars. A total of 20 cells were examined for each colony. (**D**) RT-PCR analysis of mRNA coding for endogenous proteins such as REX-1, ALP, FGF-5, FUT9, and GAPDH in NSC-like colonies (NSC), EpiSC colonies (EpiSC), human cervical carcinoma cell line HeLa (HeLa), human ovarian carcinoma cell line PA-1 (PA-1), and human skin fibroblasts HDFa (HDFa). M, 100-bp ladder markers.

**Figure 2 ijms-20-01651-f002:**
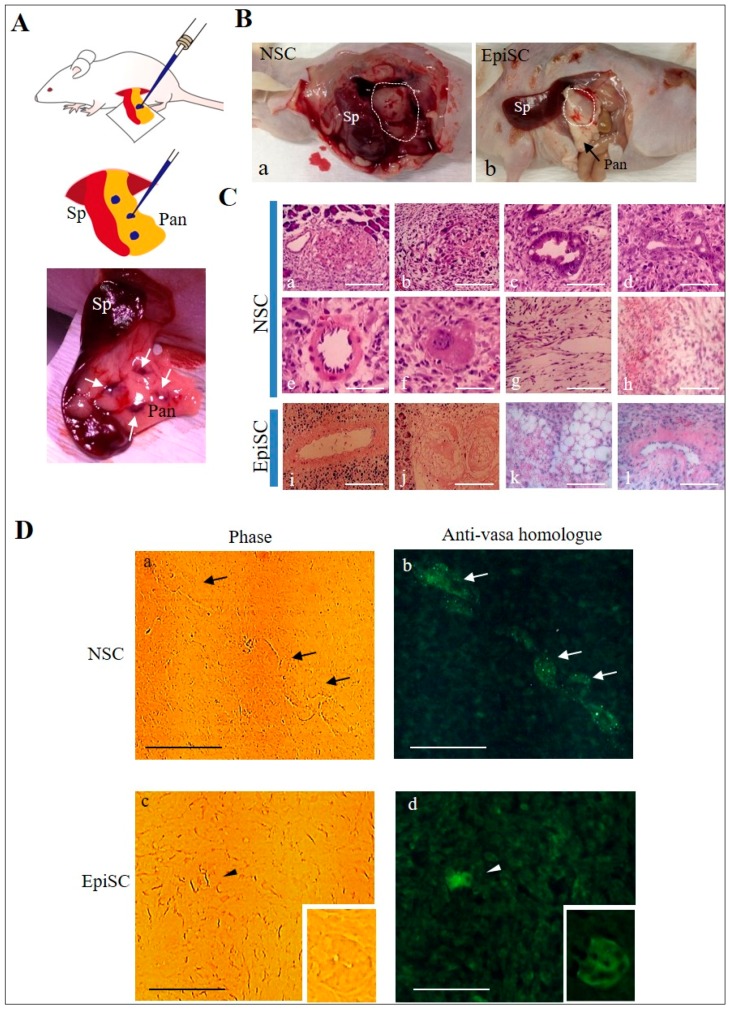
Histological analysis of teratomas generated after grafting of cultured iPSCs into the pancreatic parenchyma of BALB/c-*nu/nu* mice. (**A**) Schematic diagram of the surgery. After anesthesia, spleen, and pancreas are exposed under observation using a dissecting microscope. Then, a glass micropipette in which a solution containing iPS colonies and trypan blue (as a visible marker) is present, is inserted into the pancreatic parenchyma and immediately colonies are injected. In the bottom of the panel, four injected sites (indicated by arrows) are visible after injection. Pan, pancreas; Sp, spleen. (**B**) Solid tumors (shown by dotted lines) grown in BALB/c-*nu/nu* mice 1 (**a**; derived from inoculation of NSC-like colonies) and 1.5 (**b**; derived from inoculation of EpiSC colonies) months after grafting. Pan, pancreas; Sp, spleen. (**C**) H-E-stained sections of solid tumors generated after grafting of NSC-like colonies (**a**–**h**) or EpiSC colonies (**i**–**l**). Bar = 50 μm. (**D**) Immunostaining of solid tumors generated after grafting of NSC-like colonies (**a**,**b**) or EpiSC colonies (**c**,**d**) by anti-vasa homolog. Note that cell aggregates with structural integrity are positive for staining with anti-vasa homolog (black arrows in **a** and white arrows in **b**) in the NSC-like colony-derived tumors, but only a slight portion where no cellular integrity is present was positive for this antibody in the EpiSC colony-derived tumors (black arrowhead in **c** and white arrowhead in **d**). Cell aggregates with probably embryonic renal structure are found to be positive to the antibody (see the column on the right bottom). Bar = 100 μm.

**Figure 3 ijms-20-01651-f003:**
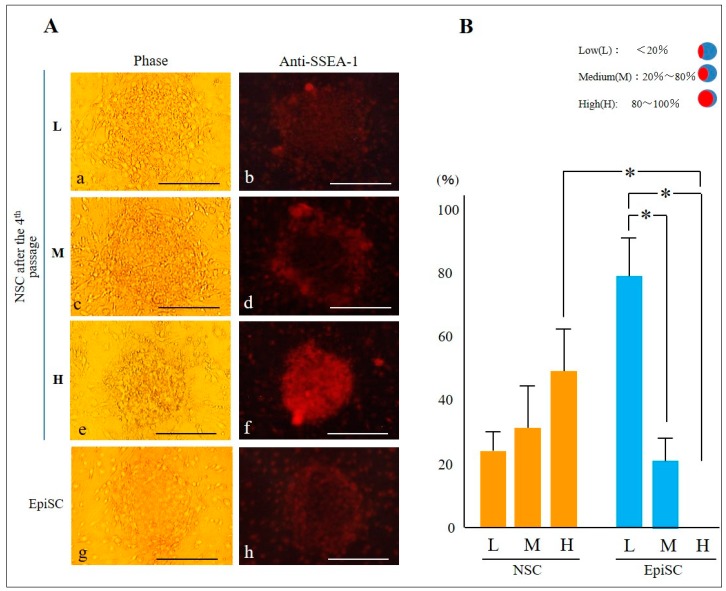
Anti-SSEA-1 can preferentially react with NSC-like colonies. (**A**) Immunostaining of NSC-like (**a–f**) and EpiSC colonies (**g,h**) by anti-SSEA-1 (Lot no. GR38347-10). Colony in a,b is classified into “L” group [showing low (L) level expression of SSEA-1], since less than 20% of an entire surface of a colony is stained by anti-SSEA-1. Similarly, the colonies in c,d or e,f are classified into “M” [showing medium (M) level expression of SSEA-1] or “H” [showing high (H) level expression of SSEA-1] group, since 20 to 80% or > 80% of an entire surface of a colony is stained by anti-SSEA-1, respectively. Phase, photos were taken under light; anti-SSEA-1, photos taken under UV illumination. Bar = 200 μm. (**B**) Summary of anti-SSEA-1-stained colonies (shown in **A**) presented as graphs. In each group, 26 colonies are examined and classified into L, M, and H groups. * indicates the statistical significance (*p* < 0.05) between the two groups.

**Table 1 ijms-20-01651-t001:** Primer sets used for RT-PCR analysis.

Gene	Forward Primer (5′–3′)	Reverse Primer (5′–3′)	Size (bp)	Reference
*REX-1*	CAGATCCTAAACAGCTCGCAGAAT	GCGTACGCAAATTAAAGTCCAGA	306	[41]
*ALP*	TGGCCCCCATGCTGAGTGACAC	TGGCGCAGGGGCACAGCAGAC	160	[35]
*FGF-5*	ATCCCACGAAGCCAATATGT	GCAGAAAGGGGAATCTTTGAC	420	GenBank M37825.1
*FUT9*	TCTACGTGCTTTCCATGATAT	CAGAGCTGGCTGATTCCATTG	180	NCBI NM_006581
*GAPDH*	GCACCGTCAAGGCTGAGAAC	TGGTGAAGACGCCAGTGGA	138	[41]

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
