# Peer review of "Increased Expression of Cell Surface SSEA-1 is Closely Associated with Naïve-Like Conversion from Human Deciduous Teeth Dental Pulp Cells-Derived iPS Cells"

_ijms, 2019, doi:10.3390/ijms20071651_

Reviewer 1 Report

The authors claim that increased expression of SSEA-1 is closely associated with naïve-like conversion of human deciduous teeth dental pulp cells to iPS cells. This is interesting report but the concept is not very new and many part of the results need to be clarified more because some results are contradictory to other researchers’ reports.

1.    In Figure 1D, marker gene expressions in HDDPC (before conversion) should be added. It is better to include Oct4 and Nanog. ALP staining result will be beneficial.

2.    In Figure 3, if the authors claim the importance of SSEA-1 expression in naïve state, expression of other typical pluripotency markers such as Oct4 and Nanog should be compared with SSEA-1 expression. Error bars should be added in Figure 3B.

3.    According to many other articles, SSEA-1 is negative in naïve state human iPSCs and this is different from murine counterpart. So additional data related to SSEA-1 are required to flip current theory.

4.    Images of Figure 2D a, c need to be replaced with clearer ones. In addition, background fluorescence in EpiSC is too strong to confirm the colony.

Author Response

1st reviewer

Comments and Suggestions for Authors

The authors claim that increased expression of SSEA-1 is closely associated with naïve-like conversion of human deciduous teeth dental pulp cells to iPS cells. This is interesting report but the concept is not very new and many part of the results need to be clarified more because some results are contradictory to other researchers’ reports.

1.    In Figure 1D, marker gene expressions in HDDPC (before conversion) should be added. It is better to include Oct4 and Nanog. ALP staining result will be beneficial.

Answer-1: Concerning the stemness marker expression of HDDPCs (from which EpiSCs have been generated), we have already examined this. According to our previous paper (Inada et al., J. Investig. Clin. Dent. 2017, 8, e12236), HDDPCs expressed OCT4 and NANOG, as well as ALP. Thus, we considered that HDDPCs contain immature cells (probably stem cells) together with more differentiated cells such as fibroblasts. This point is mentioned in the revised text (see L374-375).

2.    In Figure 3, if the authors claim the importance of SSEA-1 expression in naïve state, expression of other typical pluripotency markers such as Oct4 and Nanog should be compared with SSEA-1 expression. Error bars should be added in Figure 3B.

Answer-2: We already examined mRNA expression of OCT4 and NANOG in both EpiSCs and NSCs by RT-PCR and found no significant difference in the amounts of those molecules expressed between EpiSCs and NSCs (data not shown). Furthermore, we found that the NSC-like colonies expressed OCT4, NANOG, Tra-1-60, and SSEA-4 (See suppl. Fig. 2), all of which were also observed in the EpiSCs, as previously shown by others (see L298-299 in the revised text). Thus, we consider that the levels of these stemness factors may not be strictly correlated with the transition from epi-stem cell state to naïve cell state.      

As for the addition of error bars in Fig. 3B, we improved it in the revised Fig. 3B.

3.  According to many other articles, SSEA-1 is negative in naïve state human iPSCs and this is different from murine counterpart. So additional data related to SSEA-1 are required to flip current theory.

Answer-3: To our knowledge, human iPSCs (which are considered EpiSCs, but not NSCs) do not express SSEA-1, but in contrast mouse iPSCs (which are considered as NSCs) did. This point is already mentioned in our previous manuscript (corresponding to L75-77 in the revised manuscript). However, during transition from epi-stem cell state to naïve cell state, we found that SSEA-1 began to be expressed (which is the major point to be stressed in this paper).

As for additional data related to SSEA-1, we observed increased expression of FUT9, an enzyme required for synthesis of a fucose-based carbohydrate epitope that is specifically recognized by anti-SSEA-1, in the NSCs (see Fig. 1D). We also confirmed the presence of such epitope expression on the NSCs through cytochemical staining using rBC2LCN lectin, which also recognizes the same epitope that is recognized by anti-SSEA-1 (see Suppl. Fig. 3D_c). These data support our present finding that NSCs can express SSEA-1 on their cell surface. Furthermore, the positive staining performed by anti-SSEA-1 and rBC2LCN lectin can be abolished by co-incubation with those reagents with fucose (see Suppl. Fig. 3D_f). This indicates specificity of the reagents used here.

4.  Images of Figure 2D a, c need to be replaced with clearer ones. In addition, background fluorescence in EpiSC is too strong to confirm the colony.

Answer-4: In response to the referee’s suggestion, we improved the quality of photos in Fig. 2D-a,c and Fig. 3A-g,h (see the revised Fig. 2D and 3A).

Reviewer 2 Report

Dear authors

This is an interesting and well performed study. However, some aspects need to be improved for futher evaluations. Please consider the following comments

1)     Introduction: the authors state that “limited differentiation capability of EpiSCs makes them an inconvenient choice for use in regenerative medicine… unable to differentiate into the types of cells the researchers aim for.” although the iPSC are not the perfect cell model for regeneration, they can differentiate into several types of specialized cells. This statement is somewhat unfair. Please rephrase as the “unable to differentiate” is not a 100% true statement since the iPSC have great differentiation potential that can be used for regeneration, drug discovery and disease modeling

2)     The aim of this work is on the methods to “derive” iPSC. It would be interesting to highlight the actual achievements on the use of iPSC in the dental field. The first statement summarizes that advances were done but provides no information of how this type of cell can be useful. Please consider adding the key findings from:

-        Generation of tooth-like structures from integration-free human urine induced pluripotent stem cells

-        Functional Odontoblastic-Like Cells Derived from Human iPSCs

-        Differentiation of mouse iPS cells into ameloblast-like cells in cultures using medium conditioned by epithelial cell rests of Malassez and gelatin-coated dishes

3)     Rephrase the last paragraph. The objective and hypothesis need to be clear. Ending the introduction with a statement about the methodology is strange.

Methods

4)     It is not clear how the iPSC were generated. Please describe the methodology. The text jumps from ethical concerns to maintance…

Results

5)     “Interestingly, the average diameter of each nucleus of the cells in the dome-like colonies” how did the authors ensure that individual nucleus was being measured if the cells were overlaying? Please clarify in the text.

6)     Caption fig 3: please describe what is L, M and H. The text is long and difficult to the reader to look for the information.

Discussion

7)     The first statement is about “generation of iPSC” but the authors did not describe the methods they used to generate the ipcs. After describing properly this in the methods, please discuss briefly the advantages/disadvantages of the methods used plus the highlighted in the first part of the discussion. Further information can be found in the paper below

-        Inducing pluripotency for disease modeling, drug development and craniofacial applications

8)     The part: “we found the following: 1)…” needs to be drastically improved. Findings need to be reported and discussed and not mentioned in a “informal way” as “this is what we found”. This part is poorly written and does not reflect that high quality of the scientific content of this paper. Please revise that paragraph fully [from 1) to 5)].

9)     “NSC-like colonies tended to detach from the dish” please provide some plausible explanations for this phenomenon. Could the use of feeder or coating the plates decrease this problem? Mention advantages and disadvantages of alternative methods to solve this issue.

10)  The authors state that “SSEA-1 is closely associated with NSC (but not EpiSC)” but several publications (as below) correlate the expression of SSEA to iPSC. Please highlight that, although this was observed for this paper other publications have observed otherwise

-        Induced Pluripotent Stem Cell Clones Reprogrammed via Recombinant Adeno-Associated Virus-Mediated Transduction Contain Integrated Vector Sequences

-        Generation of a pig induced pluripotent stem cell (piPSC) line from embryonic fibroblasts by incorporating LIN28 to the four transcriptional factor-mediated reprogramming: VSMUi001-D

11)  “Although they did not point out that those SSEA-1-positive cells are NSC, we consider that they may be iPSCs in the naïve state.” The part “we consider” is completely unnecessary as the authors do not have access to their data. It is merely an assumption or personal opinion.

Author Response

2nd reviewer

Comments and Suggestions for Authors

Dear authors 

This is an interesting and well performed study. However, some aspects need to be improved for futher evaluations. Please consider the following comments

 Introduction

1) Introduction: the authors state that “limited differentiation capability of EpiSCs makes them an inconvenient choice for use in regenerative medicine… unable to differentiate into the types of cells the researchers aim for.” although the iPSC are not the perfect cell model for regeneration, they can differentiate into several types of specialized cells. This statement is somewhat unfair. Please rephrase as the “unable to differentiate” is not a 100% true statement since the iPSC have great differentiation potential that can be used for regeneration, drug discovery and disease modeling

Answer-1: As suggested by the referee, the statement “This limited differentiation capability of EpiSCs makes them an inconvenient choice for use in regenerative medicine, since they are unable to differentiate into the types of cells the researchers aim for” was a little bit overstated. We replaced the previous sentence with the following ones: “In this context, human iPSCs have the potential to differentiate into several types of specialized cells that are derived from the three germ layers, as mentioned above, but they are still not the perfect cell model for regeneration. If human iPSCs are successfully converted to NSCs, the resultant cells could be used for various studies in addition to regenerative medicine in the dental field” (see L52-55 in the revised text).

2) The aim of this work is on the methods to “derive” iPSC. It would be interesting to highlight the actual achievements on the use of iPSC in the dental field. The first statement summarizes that advances were done but provides no information of how this type of cell can be useful. Please consider adding the key findings from:

-        Generation of tooth-like structures from integration-free human urine induced pluripotent stem cells

-        Functional Odontoblastic-Like Cells Derived from Human iPSCs

-        Differentiation of mouse iPS cells into ameloblast-like cells in cultures using medium conditioned by epithelial cell rests of Malassez and gelatin-coated dishes

Answer-2: That points the referee make are helpful suggestions. Hence, we added additional information of past achievements concerning the possible application of iPSCs to the dental field in the revised text (see L40-48).  

3) Rephrase the last paragraph. The objective and hypothesis need to be clear. Ending the introduction with a statement about the methodology is strange.

Answer-3: According the referee’s suggestion, we deleted the last sentence “For induction of NSCs, we employed the method of Hana et al. who used a cocktail (2i + kenpaullone + forskolin).” from the previous manuscript. Please see L97 in the revised text. We believe this statement makes our purpose and hypothesis clear.

4)  It is not clear how the iPSC were generated. Please describe the methodology. The text jumps from ethical concerns to maintance…

Answer-4: According the referee’s suggestion, we provided a methodological description on the preparation of iPSCs (see L373-388 in the revised text).

5)  “Interestingly, the average diameter of each nucleus of the cells in the dome-like colonies” how did the authors ensure that individual nucleus was being measured if the cells were overlaying? Please clarify in the text.

Answer-5: According the referee’s suggestion, we added the following sentence: “We confirmed that there was no overlapping among DAPI-stained nuclei by measuring their dimeter after preparation of digital images of individual nuclei, based on the 3D conversion software” (see L116-118 in the revised text).

6) Caption fig 3: please describe what is L, M and H. The text is long and difficult to the reader to look for the information.

Answer-6: According the referee’s suggestion, we added statements explaining the terms L, M, and H (shown in Fig. 3) to the figure legends of Fig. 3 in the revised text.

7) The first statement is about “generation of iPSC” but the authors did not describe the methods they used to generate the ipcs. After describing properly this in the methods, please discuss briefly the advantages/disadvantages of the methods used plus the highlighted in the first part of the discussion. Further information can be found in the paper below

-        Inducing pluripotency for disease modeling, drug development and craniofacial applications

Answer-7: According the referee’s suggestion, we added information concerning the preparation of iPSCs to the revised text (see L373-390).

As for the advantages/disadvantages of the methods for induction of NSCs, we added some information about this in the revised text (see L283-293).

8) The part: “we found the following: 1)…” needs to be drastically improved. Findings need to be reported and discussed and not mentioned in a “informal way” as “this is what we found”. This part is poorly written and does not reflect that high quality of the scientific content of this paper. Please revise that paragraph fully [from 1) to 5)].

Answer-8: We improved the expression as follows: “Based on these results, it can be concluded that EpiSCs obtained after culture in NSC medium are judged as those successfully converted to NSCs” (see L308-310 in the revised text).

9) “NSC-like colonies tended to detach from the dish” please provide some plausible explanations for this phenomenon. Could the use of feeder or coating the plates decrease this problem? Mention advantages and disadvantages of alternative methods to solve this issue.

Answer-9: We always seeded iPSCs on the feeder cells in a gelatin-coated dish to maintain them in vitro, which is generally employed as standard cultivation of iPSCs. Even with these conditions, the NSC-like colonies tend to be detached from the dish. It remains unknown why such an event frequently occurs. We have provided an additional sentence explaining this event as follows: “It remains unknown why such an event frequently occurs, given that we always seeded iPSCs on the feeder cells in a gelatin-coated dish to maintain them in vitro, which is a technique generally employed as part of the standard cultivation of iPSCs. Likely, the property that allows NSC-like colonies to attach to the feeder cells or substratum may have been modified during culture” (see L312-316 in the revised text).

10)  The authors state that “SSEA-1 is closely associated with NSC (but not EpiSC)” but several publications (as below) correlate the expression of SSEA to iPSC. Please highlight that, although this was observed for this paper other publications have observed otherwise

-        Induced Pluripotent Stem Cell Clones Reprogrammed via Recombinant Adeno-Associated Virus-Mediated Transduction Contain Integrated Vector Sequences

-        Generation of a pig induced pluripotent stem cell (piPSC) line from embryonic fibroblasts by incorporating LIN28 to the four transcriptional factor-mediated reprogramming: VSMUi001-D

Answer-10: The papers the referee recommends dealt with murine and porcine iPSCs but not with human iPSCs. Based on the referee’s suggestion, in the revised text, we added the following sentence: “As mentioned previously, SSEA-1 expression is known to occur in murine iPSCs/ESCs, but not in human iPSCs/ESCs. Here we found that SSEA-1 is expressed when human iPSCs are converted to the naïve state, which was verified by immunostaining with anti-SSEA-1” (see L336-339 in the revised text).

11)  “Although they did not point out that those SSEA-1-positive cells are NSC, we consider that they may be iPSCs in the naïve state.” The part “we consider” is completely unnecessary as the authors do not have access to their data. It is merely an assumption or personal opinion.

Answer-11: As suggested, we improved this as follows: “Although the authors did not specify that the SSEA-1-positive cells are NSCs, it is possible that they are iPSCs in the naïve state” (see L353-354 in the revised text). 

Round  2

Reviewer 1 Report

Accepted because the authors revised the manuscript accordingly.

Reviewer 2 Report

This version is much improved